# Space-Based Displacement Monitoring of Coastal Urban Areas: The Case of Limassol's Coastal Front

Kyriaki Fotiou [1,2,*], Dimitris Kakoullis [1,2], Marina Pekri [1,2], George Melillos [1,2], Ramon Brcic [3], Michael Eineder [3], Diofantos G. Hadjimitsis [1,2] and Chris Danezis [1,2]

1   Department of Civil Engineering and Geomatics, Cyprus University of Technology, P.O. Box 50329, Limassol 3036, Cyprus; dimitris.kakoullis@cut.ac.cy (D.K.); marina.pekri@cut.ac.cy (M.P.); georgios.n.melillos@cut.ac.cy (G.M.); d.hadjimitsis@cut.ac.cy (D.G.H.); chris.danezis@cut.ac.cy (C.D.)
2   Eratosthenes Centre of Excellence, Limassol 3036, Cyprus
3   German Aerospace Center, Remote Sensing Technology Institute, Oberpfaffenhofen, 82234 Weßling, Germany; ramon.brcic@dlr.de (R.B.); michael.eineder@dlr.de (M.E.)
*   Correspondence: kyriaki.fotiou@cut.ac.cy; Tel.: +357-25245014

**Abstract:** In the last five years, the urban development of the city of Limassol has rapidly increased in the sectors of industry, trade, real estate, and many others. This exponentially increased urban development arises several concerns about the aggravation of potential land subsidence in the Limassol coastal front. Forty six Copernicus Sentinel-1 acquisitions from 2017 to 2021 have been processed and analyzed using the Sentinel Application Platform (SNAP) and the Stanford Method for Persistent Scatterers (StaMPS). A case study for the identification and analysis of the persistent scatterers (PS) in pixels in a series of interferograms and the quantity of the land displacements in the line of sight of the Limassol coastal front is presented in this research, with subsidence rates up to about ($-5$ to 4 mm/year). For the validation of the detected deformation, accurate ground-based geodetic measurements along the coastal area were used. Concordantly, considering that there is a significant number of skyscrapers planned or currently under construction, this study attempts a preliminary assessment of the impact these structures will pose on the coastal front of the area of Limassol.

**Keywords:** displacement monitoring; land subsidence; coastal areas; PSI; SAR; leveling; Cyprus

## 1. Introduction

The notable urban and infrastructure growth of the last five years in Limassol, Cyprus, highlighted the necessity to carry out a detailed investigation on potential land subsidence in the coastal zone. Following the crisis events of 2013 [1], the Cyprus Government promoted incentives for land development to revive the Cyprus economy. Consequently, Limassol became the fastest-growing city in Cyprus in construction, with skyscrapers and tall buildings, built one after the other along the coastal front, on a stretch of almost 20 km. This massive development attracted foreign investments, generating increased concern that a combination of factors, such as overexploitation of groundwater, the structures' load, the sea level rise driven by global climate change, and the intense earthquake activity, may holistically trigger land subsidence phenomena with severe impacts [2]. Land subsidence can be defined as the differential sinking of the ground surface with respect to the surrounding terrain or sea level. The later poses an imminent threat for the socioeconomic equilibrium of the country, as well as an incremental factor for the risk of possible floods in the specific area [3].

The integrated use of multiple space-borne InSAR (Interferometric Synthetic Aperture Radar) techniques is, undoubtedly, among the most effective and accurate methods to monitor land subsidence [4] and, therefore, assess the impact of urban infrastructures on the coastal zones [5]. Techniques such as Persistent Scatterer Interferometry (PSI) presented

by Feretti et.al [6] and Differential Interferometry (D-InSAR) and Small Baseline Subset Interferometry SBAS [7] became indispensable parts in ground deformation monitoring analysis of urban areas as they may provide cm- to mm-level-accuracy products [8]. Investigating the land subsidence in an urban environment using D-InSAR may be inadequate due to the lack of coherence in multitemporal images in cities, caused by temporal–geometrical decorrelations [9,10] and atmospheric inhomogeneities [11]. Thus, the use of PSI could be characterized as inevitable. The PSI technique detects and measures specific points (buildings, stable rocks, roads, etc.) on the surface of the Earth that are phase-coherent and stable over a period of time [12,13]. The displacements are only measured in the direction of the axis connecting the target and sensor (LoS); this axis has a different orientation in space for ascending and descending orbits [14]. Only descending orbits are considered in the current research, since the ascending ones were affected by various illumination effects arising from SAR images' side-looking scene illumination. Specifically, in densely built-up areas with tall buildings, large portions of the data can be interfered with by the shadowing, layover, and foreshortening effects due to the slope orientation, geometry, and height of the buildings in an urban environment, as well as the double-bouncing phenomenon of the radar signal [15].

A plethora of studies has proved the ability of land subsidence monitoring in coastal urban areas in many nations worldwide, combining different space-based and ground-based techniques, such as PSI and SBAS and GNSS, using various satellite constellations and leveling techniques. As shown in Table 1, Sentinel-1, provided by the ESA satellite mission, has been most usable technology in the past few years, as long as the PSI technique is the most applicable for quantifying land subsidence in urban cities. Specifically, according to the geology of cities, almost all locations are faced with land subsidence worldwide, consisting of alluvial soil, and the subsidence rates are ranged between mm and cm/year, depending on their development status and the years of the study. Utilizing technology and analyzing the findings so far on these landslides, it can be relatively concluded that many coastal cities are highly affected and may even be at high risk in the future.

**Table 1.** Previous studies and estimated land subsidence rates in coastal areas worldwide, combining space- and ground-based techniques.

| Previous Study | Data/Technique | Subsidence Rate |
|---|---|---|
| Mancini F. et al. [16] | Sentinel-1A,B/PSI from 2015 to 2019 | Avg: −2 mm/year |
| Cigna F. et al. [17] | Sentinel-1/PSI, SBAS, leveling from 2014 to 2016 | Avg: −8 mm/year |
| Darwish N. et al. [18] | Sentinel-1, ALOS-PALSAR-2/SBAS from 2015 to 2020 | Avg: −12.5 mm/year |
| Bedini E. et al. [19] | Sentinel-1/PSI from 2017 to 2018 | Avg: −30 mm/year |
| Imamoglu M. et al. [20] | Sentinel-1/PSI from 2014 to 2018 | Avg: −35 mm/year |
| Radeb A. et al. [21] | Sentinel-1/SBAS from 2015 to 2019 | Avg: −16 mm/year |
| Aslan G. et al. [22] | Sentinel-1/PSI from 2014 to 2017 | Avg: −10 mm/yr |
| Tosi L. et al. [23] | TerraSAR-X/PSI from 2008 to 2013 | Avg:+1–4 mm/year |
| Nur A.S. et al. [24] | Sentinel-1/PSI from 2017 to 2020 | Avg: −20.95 mm/year |
| Hakim W. et al. [25] | Sentinel-1/PSI from 2017 to 2020 | Avg: −35 mm/year |
| Gido N.A.A. et al. [26] | Sentinel-1/PSI, leveling 2015 and May 2020 | Avg: −2 to −6 mm/year |
| Delgado Blasco J.M. et al. [27] | Sentinel-1/PSI from 2015 to 2018 | Avg: −20 to 2 mm/year |
| Hassan S.R. et al. [28] | Sentinel-1/PSI from 2014 to 2019 | Avg: +2–24 mm/year |
| Hu B. et al. [29] | Sentinel-1/PSI, SBAS and GNSS year 2016 | Avg: −20 to 10 mm/year |
| Aimaiti Y. et al. [30] | ERS-1/-2, ALOS PALSAR/SBAS, leveling from 1993 to 2010 and 2014 to 2017 | Avg: −28 mm/year and −18 mm/year |
| Alatza S. et al. [31] | Sentinel-1, Envisat/PSI, SBAS, GPS from 2003 to 2019 | Avg: +5 mm/year |

**Table 1.** *Cont.*

| Previous Study | Data/Technique | Subsidence Rate |
| --- | --- | --- |
| Fiaschi S. et al. [32] | Sentinel-1/PSI from 2015 to March 2018 | Avg: −17 mm/year |
| Cigna F. et al. [33] | Sentinel-1/PSI, P-SBAS from 2014 to 2020 | Avg: −2.3 cm/year |
| Cian F. et.al. [34] | Sentinel-1, TerraSAR, COSMO-SkyMed, Envisar-ASAR/PSI from 2004 to 2018 | Avg: −9 to 5 mm/year |
| Duffy C.E. et al. [35] | Sentinel-1/PSI from 2017 to 2019 | Avg: −3.3 mm per year |

Using space-based deformation monitoring techniques to investigate the land subsidence occurring along the Limassol coastal front, the purpose of this work is two-fold. As the first objective, this research focuses on quantifying the land subsidence of the area using remote sensing techniques and validating the detected deformation, if any, using conventional in-situ ground-based data. As a secondary objective, this research investigates the possibility of automatically monitoring land subsidence in urban coastal areas, consisting of skyscrapers and tall buildings, through space-based techniques. Concordantly, taking into account that there is a significant number of skyscrapers that are in the planning or developmental stages, this study attempts a preliminary assessment, characteristics, and discussion of the impact these structures will pose on the coastal front of the area of Limassol through the Cyprus Continuously Operating Natural Hazards Monitoring and Prevention System (CyCLOPS) strategic research infrastructure unit [36].

## 2. Materials and Methods

### 2.1. Case Study—Limassol Coastal Front

Limassol is found on the south part of the island and is the second-largest urban city in Cyprus after Nicosia, covering an area of almost 53 km². The Port of Limassol is one of the busiest ports in the Mediterranean transit trade and generally the largest port in Cyprus. Furthermore, Limassol is the base of Cyprus University of Technology, one of three state universities, established in 2004. The coastal front of Limassol City encompasses an area of about 20 km² [Figure 1]. The area of interest is relatively flat, with a height that reaches almost 1.5 m above mean sea level. From a geological point of view, as occurs in the majority of coastal cities, the geology of the Limassol coastal zone consists of alluvial soils.

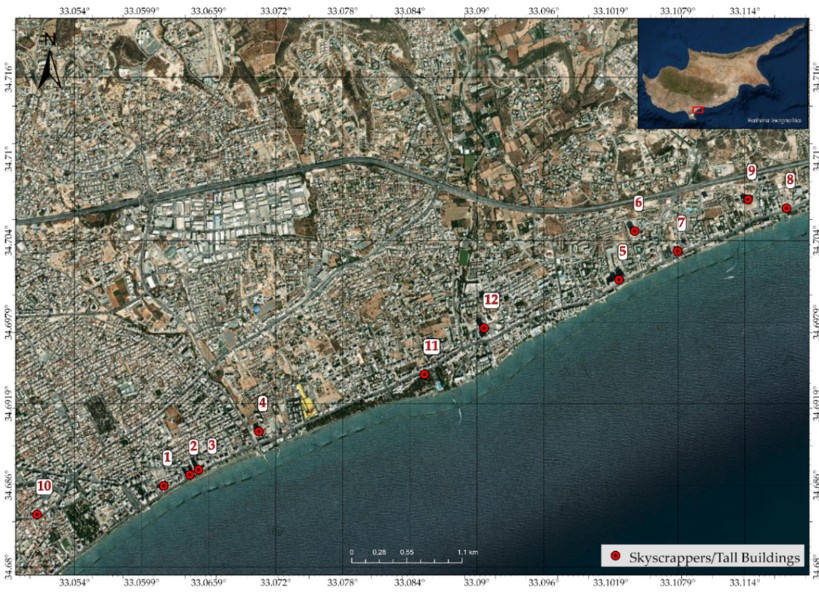

**Figure 1.** Case study of Limassol coastal front in Cyprus with newly built skyscrapers and tall buildings.

As of late 2013, Cyprus has been experiencing a construction regeneration, and Limassol is the focal point, upgrading the image of the city as a cosmopolitan and attractive destination. Since 2013, when legislation was passed encouraging construction, a significant number of skyscrapers and tall buildings have been introduced to the Cypriot community every year. The construction industry in Cyprus counts for 7% of the country's GDP. To date, there are about 70 buildings that have been proposed and/or are under construction that will stand taller than 50 m when completed, of which 29 are skyscrapers. Specifically, Table 2 presents, in detail, the name of the projects, their height, and the number of floors, as well as the year of their construction and their exact locations in WGS1984 coordinate system (EPSG4326). A recently completed residential high-rise building, One, is the tallest tower in Cyprus, the tallest seafront residential building in Europe, and the 48th-tallest building in the European Union. Among others, one current under-construction project is the 'City of Dreams Mediterranean,' Europe's largest casino resort.

**Table 2.** Features of buildings on Limassol coastal front, Cyprus.

| Rank | Name | Height (m) | Floors | Construction Year | Location | Image |
|------|------|-----------|--------|-------------------|----------|-------|
| 1 | Trilogy | 161 | 39 | 2022 | 33.06196° E 34.68596°N |  |
| 2 | Olympic Residence | 76 | 20 | 2012 | 33.06426° E 34.68679° N |  |
| 3 | One | 170 | 38 | 2021 | 33.06511° E 34.68709° N |  |
| 4 | The Oval | 75 | 17 | 2017 | 33.07052° E 34.68992° N |  |
| 5 | Limassol Del Mar | 107 | 27 | 2021 | 33.10273° E 34.70108° N |  |
| 6 | Sky Tower | 103 | 24 | 2021 | 33.10416° E 34.70463° N |  |
| 7 | Arc Ship Tower | 67 | 17 | 2018 | 33.10798° E 34.70312° N |  |

**Table 2.** *Cont.*

| Rank | Name | Height (m) | Floors | Construction Year | Location | Image |
|------|------|------------|--------|-------------------|----------|-------|
| 8 | Dream Tower | 119 | 27 | 2023 | 33.11762° E 34.70637° N |  |
| 9 | iHome | 56 | 11 | 2018 | 33.11431° E 34.70687° N |  |
| 10 | The Icon | 104 | 21 | 2021 | 33.05059° E 34.68385° N |  |
| 11 | MARR Tower | 116 | 24 | 2019 | 33.08530° E 34.69408° N |  |
| 12 | DTA Tower | 100 | 22 | 2019 | 33.09071° E 34.69755° N |  |

Since the island of Cyprus is located in the Mediterranean fault zone, interactions between the Eurasian and the African plates, a unique site for geodynamic analysis, are exhibited. Taking into consideration the geology of the area, the seismicity of Cyprus in the last five years [37], the sea level rise [38] due to the thermal expansion of seawater due to ocean warming and water input from land ice melt, the significant vulnerability to climate change, and the rapid construction of skyscrapers and tall buildings in the area, land subsidence monitoring is an increasingly eminent component of risk planning and decision-making regarding infrastructure. To date, no cases of land subsidence have been identified in the study area and quantified through various InSAR techniques such as PSI.

*2.2. Data Processing*

Monitoring land subsidence in the coastal front of Limassol city has been carried out using several techniques ranging from traditional leveling to PSI, as shown in Figure 2.

2.2.1. Satellite Data Processing

Considering the novelty of the research due to the lack of previous investigations in urban areas of Cyprus and the fact that a significant number of buildings are in the planning or developmental stages, open data policies through multiple hubs, which involve the use of non-commercial satellite data, were chosen in order to guarantee the reusability of the current study approach on larger spatial and time scales. More specifically, Copernicus satellites' imagery data, particularly focusing on Sentinel-1 imagery data, were used. This satellite system provides free data with global coverage with a weekly revisit period [39–41]. In conjunction with short temporal baselines, the possibility of coherent phase information between the primary and secondary images increases, thus contributing to a potentially greater density of Persistent Scatterers (PS) for estimating deformation over time [42]. The processing and analyzing of Copernicus Sentinel-1 data, covering a time interval ranging from March 2016 to March 2021, were implemented by using the Sentinel Application

Platform (SNAP), the Stanford Method for Persistent Scatterers (StaMPS/MTI) [43], and the 'snap2stamps' python workflow. StaMPS/MTI is a Sentinel-1 TOPS data PSI processing open-source software package that is demonstrated in automated mode. At the same time, 'snap2stamps' contains python scripts to automate the preprocessing of Sentinel-1 SLC data and their preparation for ingestion by StaMPS [44–46].

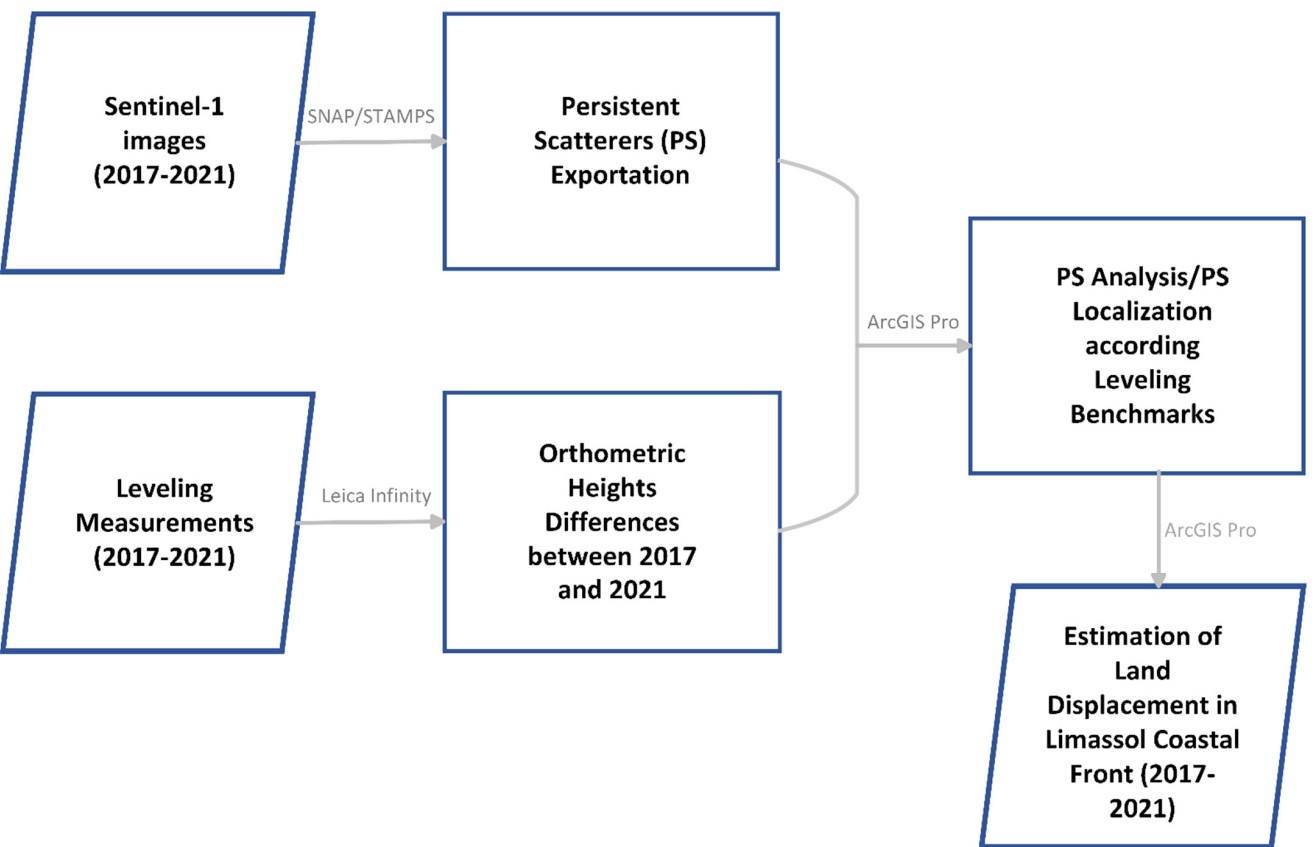

**Figure 2.** Schematic diagram presenting the workflow of the methodology combining PSI and leveling techniques for the estimation of land subsidence on the Limassol coastal front from 2017 to 2021.

Forty-six Sentinel-1B advanced synthetic aperture radar images acquired by the European Space Agency from the satellite during descending acquisition mode from 2017 to 2021 were used to identify and calculate the deformation information, as presented in Table 3, functioning in C-band (5.4 GHz). It is noteworthy that the dataset follows a repeat sequence interval of almost 30 days between every image in order to avoid decorrelation in the temporal analysis of the baseline. This period was selected for two reasons: Firstly, most of the buildings' development started in late 2016, so the earlier study will be insufficient. Secondly, the benchmarks' network for leveling reasons was designed and implemented by the Laboratory of Geodesy of the Cyprus University of Technology throughout 2017. The Sentinel-1 dataset is composed of Single-Look Complex (SLC) images with Interferometric Wide (IW) mode, with similar characteristics, creating a time series sequence over the five-year period. The Interferometric Wide (IW) swath mode, applied in the present research, acquires data with a 250 km swath, with a high spatial resolution of $5 \times 20$ m, and by using Terrain Observation with Progressive Scans SAR (TOPS), it captures one subswath in range direction [47]. Specifically, in the Sentinel-1 scenes observed by the 167 relative orbit number, the orbit passed over Limassol in descending acquisition mode. Moreover, according to Kopel et al. [48], the most favorable polarization mode for monitoring land subsidence in an urban environment is the VV polarization mode, as was used in the

dataset for this specific study since it strongly returns part of the signal to the sensor due to the artificial scatterers on the surface (e.g., buildings).

**Table 3.** Calendar of Sentinel-1 images that were used for PSI.

| Date | 2017 | 2018 | 2019 | 2020 | 2021 |
|---|---|---|---|---|---|
| January | - | 17 | 18 | 13 | 13 |
| February | - | 16 | 17 | 18 | - |
| March | - | 18 | 19 | 19 | 14 |
| April | 16 | 17 | 12 | 18 | 13 |
| May | 16 | 17 | 18 | 18 | 13 |
| June | 15 | 16 | 17 | 17 | - |
| July | - | 16 | **17** [1] | 17 | - |
| August | 20 | 15 | 16 | 16 | - |
| September | 19 | 14 | 15 | 15 | - |
| October | 19 | 20 | 15 | 15 | - |
| November | 18 | 19 | 14 | 14 | - |
| December | 18 | 19 | 14 | 14 | - |

[1] 17 July 2021 is the master image of the dataset.

### 2.2.2. SNAP-STAMPS PSI Processing

The PSI processing was divided into two independent workflows. The first workflow concerned the preprocessing of the images and the single-master DInSAR processing using ESA SNAP and 'snap2stamps', and the second workflow comprised PSI processing using StaMPS/MTI.

Initially, the image acquired on 17 July 2019 was selected and used as the 'master' image in the preprocessing procedure. The master image minimizes the sum decorrelation of all interferograms by eliminating the perpendicular baseline values and the Doppler phenomenon among all images and concordantly maximizing the expected coherence in the interferometric stack [49]. This results in the geometric and interferometric coregistration of the dataset with respect to a common frame (master image). In each of the subswaths in the preprocessing workflow, three bursts were separated in the azimuth direction. The orbital information demonstrated accurate satellite position and velocity information following the SNAP image processing. Apart from that, the geocoding of all images was applied, using the SRTM-1m [50]. Equally important was the enhancement of the spectral diversity, as well as the extraction of the topo-phase and the generation of the interferograms for each date. Subsequently, after the phase noise estimation (SNR) and the phase unwrapping, a mean velocity map of all images was computed, as shown in Figure 3.

An important omission was the conversation of phase displacements of LoS to vertical displacements. Specifically, according to Foumelis et al. [51], the conversion from the unwrapped phase to displacements are in LoS and are relative between the images. In order to calculate the vertical displacement, a formula was applied to SNAP using the band maths module including the incidence angle as presented in equation 1. Overall, as was expected, lower estimated mean velocity values were found all across the coastal line, along which the buildings are located.

$$Vertical\ Displacement = \frac{(unwrapped\ phase \cdot wavelength)}{[-4\pi \cdot \cos(incidence\ angle)]} \tag{1}$$

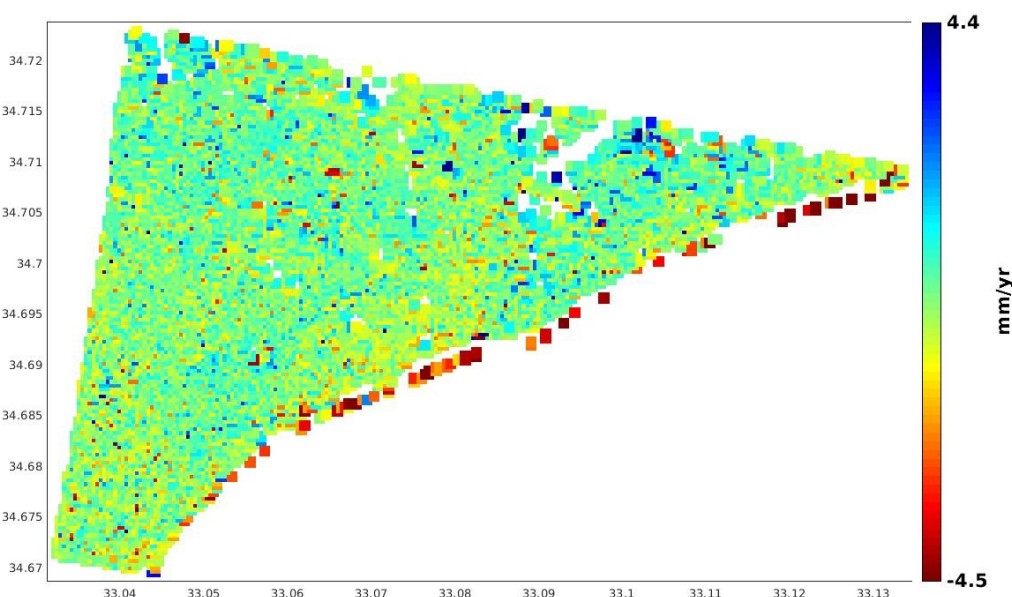

**Figure 3.** Vertical displacements of PSs estimated for period 2017–2021 generated by SNAP and STAMPS software using Sentinel-1 data.

After completing the procedure, the Persistent Scatterers (PS) were created. Specifically, the number of the final persistent scatterers was computed in three main stages. Firstly, all the candidate scatterers were identified in the study area. Persistent scatterers are pixels or points characterized by stability over time of the backscatter electromagnetic signal. Following this step, extracting the possible scatterers with a minimum value of coherence threshold equal to 0.80 was performed. Due to the density of the buildings and the small area, the coherence value should at the maximum extent possible to avoid the matching of the same PSs to more than one buildings [50]. Last but not least, the final scatterers that corresponded to tall buildings or skyscrapers in every pixel were isolated. The number of persistent scatterers is affected by various factors, including the size of the building, the material, and structures of the roof, the orientation of the building relative to radar look direction, and nearby vegetation [52]. Figure 4a represents an example of a distribution and the number of PSs within a buffer zone over a specific chosen point. At the same time, a time series graph was created, which calculated and visualized the mean velocity of all points that were within the buffer zone and plotted them for each of the forty-six interferograms (Figure 4b). Figure 5 is an excerpt of PS distribution in Google Earth Pro.

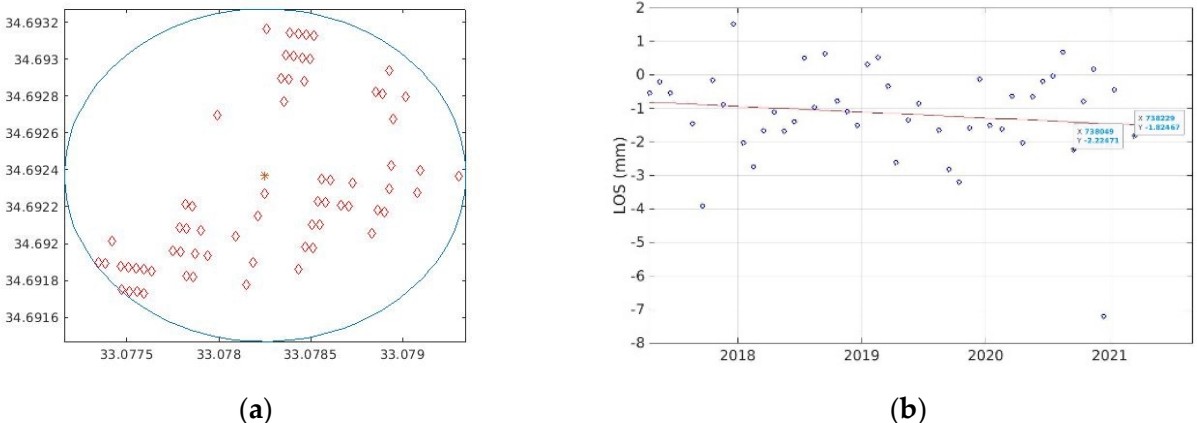

**(a)**          **(b)**

**Figure 4.** Example of (**a**) PS distribution and quantity in a 1 km buffer zone of a specific point (* defined as the center point) and (**b**) time series graph shows the mean velocity of the point computed by the different interferograms from 2017 to 2021.

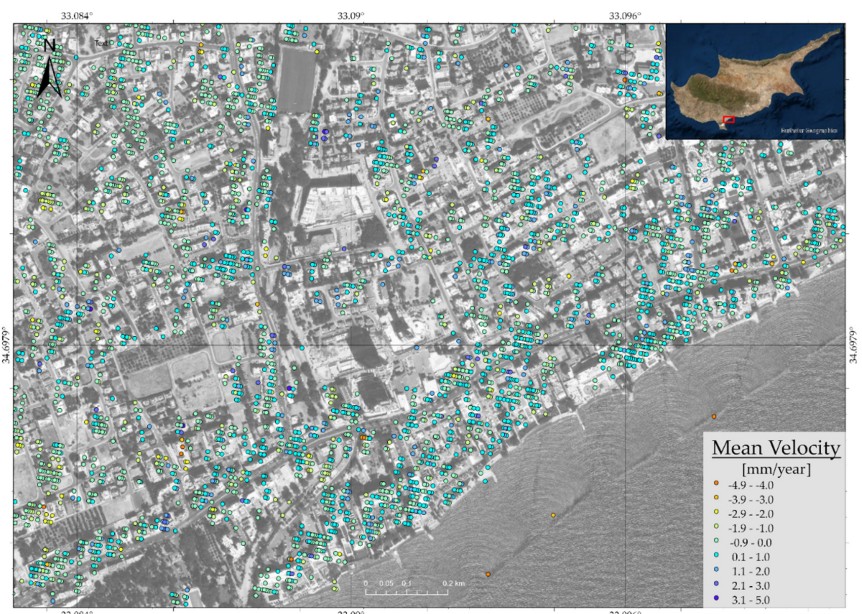

**Figure 5.** Mean velocity of persistent scatterers at the area of study (Limassol coastal front).

### 2.2.3. Leveling Equipment and Measurements

As a ground-based geodetic technique, precise leveling was selected for its simplicity and unmatched accuracy [53]. In total, thirteen (13) benchmarks across the coastline of Limassol were remeasured [Figure 6, Table 4]. The CUT Laboratory of Geodesy established a vertical control network that covers the broader Limassol area in late 2016. Since then, yearly geodetic leveling surveys have been carried out to monitor the vertical component variability. The leveling network was tied to the CUT Tide Gauge station in Limassol, which is a part of the National Geodetic Infrastructure and, therefore, tied to the national vertical datum of Cyprus. For the specific study, the leveling benchmark CUT2 was considered as our control benchmark [Figure 7]. The height difference between epochs 2017 and 2021 was used as a reference for the validation of PSI results at Limassol coastal front.

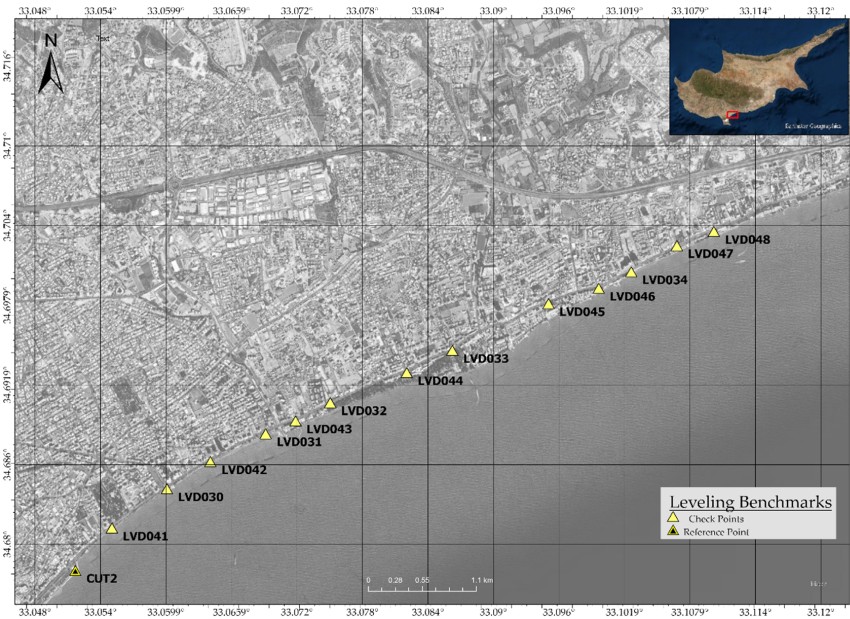

**Figure 6.** Locations of leveling benchmarks on Limassol coastal front. The yellow circle defines the control benchmark (CUT2), while yellow triangles show the measured benchmarks. All heights were obtained from the national vertical datum of Cyprus.

**Table 4.** Features (ID, height) of thirteen (13) leveling benchmarks in 2017–2021.

| Leveling Benchmark ID | Orthometric Height 2017 (m) | Orthometric Height 2021 (m) | Height Differences 2017–2021 (m) |
|---|---|---|---|
| LVD048 | 3.401 | 3.402 | 0.001 |
| LVD047 | 3.618 | 3.619 | 0.001 |
| LVD034 | 2.181 | 2.174 | −0.007 |
| LVD046 | 3.530 | 3.532 | 0.002 |
| LVD045 | 5.399 | 5.401 | 0.002 |
| LVD033 | 3.591 | 3.593 | 0.002 |
| LVD044 | 3.429 | 3.433 | 0.004 |
| LVD032 | 3.471 | 3.473 | 0.002 |
| LVD043 | 2.581 | 2.583 | 0.002 |
| LVD031 | 2.696 | 2.700 | 0.004 |
| LVD042 | 2.559 | 2.583 | 0.024 |
| LVD030 | 2.442 | 2.443 | 0.001 |
| LVD041 | 1.805 | 1.807 | 0.002 |
| **CUT2** [1] | 1.877 | 1.877 | 0.000 |

[1] CUT2 is the control benchmark.

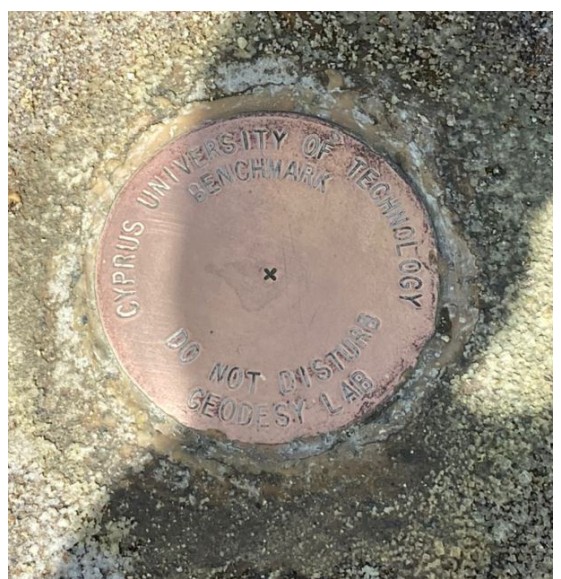 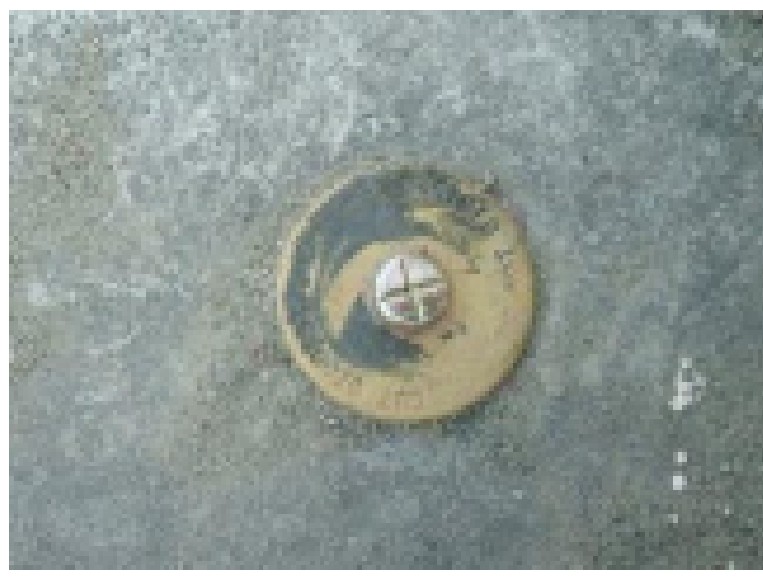

(**a**)  (**b**)

**Figure 7.** (**a**) CUT2 control leveling benchmark and (**b**) checkpoint leveling benchmarks installed by the CUT Laboratory of Geodesy in 2016.

In the current study, the height differences were acquired by leveling surveys using the high-precision digital Leica LS15 and two fiberglass staffs (GTM4L). The calculation of the orthometric height was computed in Leica Infinity software. The measurements were provided with a fiberglass bar code that allowed a resolution of 0.1 mm in electronic height measurements, offering a height accuracy of ±1.0 mm per 1 km for double-run measurements (ISO 17123-2). The procedures conducted before (instrument alignment), during (alternating back and forward sightings), and after (the orthometric correction) the measurements eliminated the main sources of errors such as rod alignment, the curvature

and refraction errors, and other several systematic errors. The tolerance error for the first-order class II survey of a section of 2 km (maximum) line length is 4 mm $\sqrt{K}$ and 5 mm $\sqrt{K}$ for an entire leveling line, where K is the length of the leveling line, measured in kilometer units [54]. It should be noted that the heights of leveling benchmarks were referenced to the Cyprus Vertical Datum.

2.2.4. Combination of the Methodologies in GIS Environment

After the data processing was completed, all information mentioned above associated with the building characteristics, PS mean velocity, displacements, and leveling measurements were imported into ESRI's ArcGIS Pro software for further analysis.

Firstly, concerning the buildings as a unit, a database was created in order to assemble all possible geographical and descriptive information about each one. Specifically, the database includes the geolocation of the building exported by Google Earth Pro. In addition to this, important information about the height and the floors of the structure was added to the database. Equally important was the starting date of the construction, as well as the ending date, in order to understand the result values of PS displacements. Apart from that, recent photos of each structure are included in the database, as well as the exact coordinates in WGS 1984, of each construction. As mentioned before, the results of this study are preliminary; hence, a well-structured and detailed database will be able to provide information for future studies over time.

As is well known, the PSI technique provides spatially scattered measurements of mm-to cm-level-accuracy displacements with spatial density depending on the resolution of each sensor. Taking into consideration the high density of PS on the infrastructures and, therefore, the validity of the PSI results, a more detailed analysis, isolating and analyzing PS separately, was carried out on the structural components. The total number of PSs extracted by STaMPS and SNAP in Limassol equals 57,916. Ergo, after investigating PSs around/on each construction, a buffer zone of 200 m that includes PSs and structure information around every building was created. The number of 200 m was used as an assumption to demonstrate the land subsidence in each building and its surrounding area through PS analysis. No interpolation method (e.g., Kriging, IDW, etc.) was used because the study focused on the structures and not the region in general.

## 3. Results

### 3.1. PSI Classification and Trends of PSI Results

Following the above methodology, the geographic information (location) of the corresponding PSs and the vertical displacements in millimeters/year and their statistics were extracted. Concerning the classification of the PSs values, it was held in ten (10) classes for visualization purposes. Each class was equally distanced by 1 mm, ranging from −5 mm to 4 mm within the studied period. For further analysis, the PSs included in a 200 m buffer zone were analyzed for each structure, and the mean value was determined for the displacement in the vertical component in each case.

As shown in Figures 4 and 5, where the location distribution of the PSs in the buffer zone is presented on an excerpt of a map, there was a significant variance in the point density in each case. Table 5 contains the quantity of PSs within the 200 m buffer zone around each construction area, as well as the construction year. This is ascribable to the starting date of each construction and the presence of the buildings in all Sentinel-1 scenes, according to the master date (17 July 2021). Thereby, the buffer zone of 200 m around each building was created, including PSs, as presented in Figure 8. For each construction, statistics data describing the mean, minimum, and maximum value of the vertical subsidence were computed. For the presentation of the PSI results, a categorization of the buildings into skyscrapers (<100 m) and tall buildings (<100 m) according to the height component [55] was carried out.

**Table 5.** Quantity of PSs within the 200 m buffer zone around each construction area and their construction year.

| Name | Construction Year | Number of PS |
|---|---|---|
| Trilogy | 2012 | 156 |
| Olympic Residence | 2018 | 214 |
| One | 2022 | 244 |
| The Oval | 2023 | 273 |
| Limassol Del Mar | 2017 | 105 |
| Sky Tower | 2021 | 145 |
| Arc Ship Tower | 2019 | 140 |
| Dream Tower | 2021 | 86 |
| iHome | 2018 | 145 |
| The Icon | 2021 | 395 |
| MARR Tower | 2019 | 144 |
| DTA Tower | 2021 | 135 |

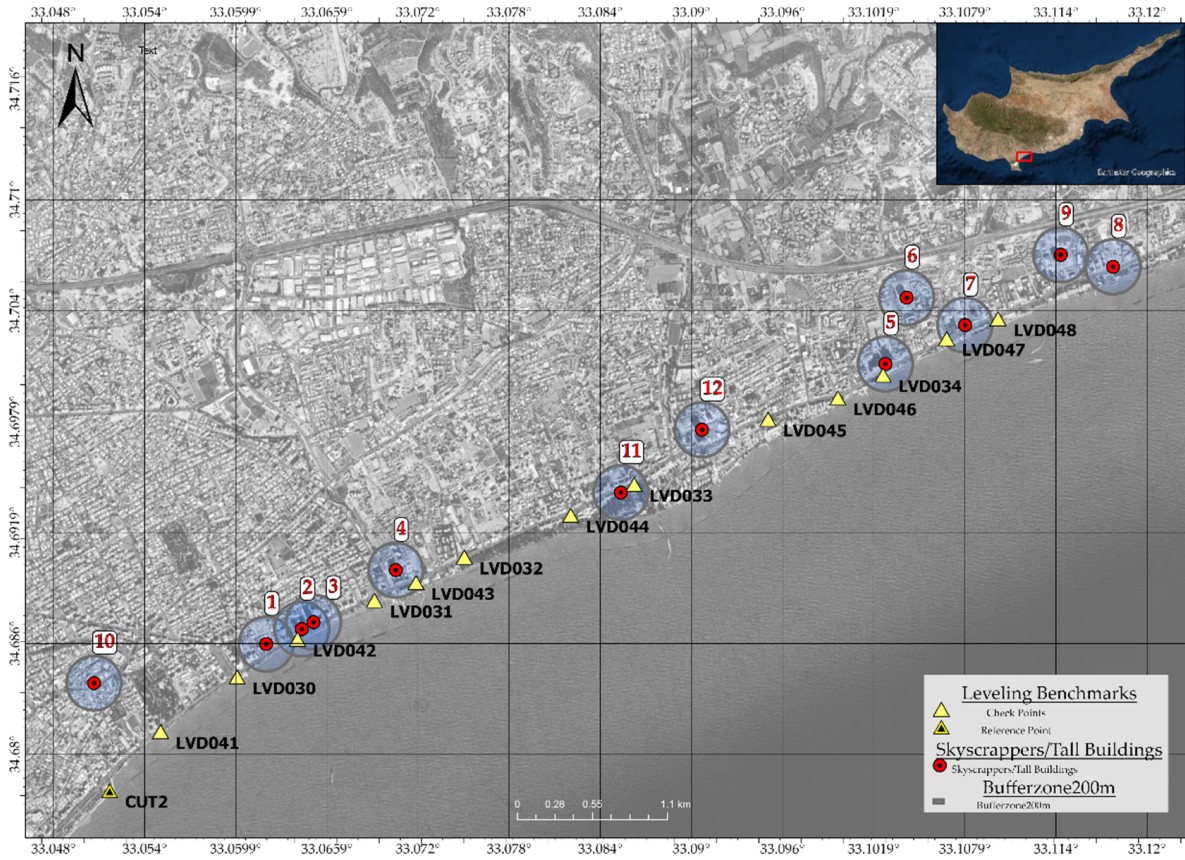

**Figure 8.** Map of skyscrapers/tall Buildings and their 200 m buffer zone with leveling benchmarks across Limassol coastal front.

3.1.1. PSI Results Analysis of Skyscrapers on Limassol Coastal Front

The skyscraper category encompasses Trilogy, The One, The Dream Tower, Limassol Del Mar, The ICON, Sky Tower, MARR Tower, and DTA Tower. As shown in Table 6, the land subsidence of the area was affected the most by the constructions in the region of Trilogy. The Trilogy construction includes three similar buildings (two facing the sea to the

south and one facing Limassol to the north) with a mean value of vertical displacement equal to −1.80 mm from 2017 to 2021. Comparably with other skyscrapers, except the building named ONE, the mean value of the Trilogy building is more than doubled. The minimum value of vertical displacement is equal to −4.51 mm. Noteworthily, the third building of the complex, facing north, is still under construction. The structures complete the list of the skyscrapers on the Limassol coastal front, showing that from the first day of their construction up to the present, the land subsidence of the built area has reached a steady state leaning towards zero.

**Table 6.** PSI statistical analysis of skyscrapers on Limassol coastal front.

| Map of PSI Analysis—Skyscrapers | Statistics (mm) | | |
| --- | --- | --- | --- |
| | Mean/Average | Maximum | Minimum |
|  | −0.944 | 2.47 | −4.51 |
|  | −0.87 | 2.43 | −4.16 |

**Table 6.** *Cont.*

| Map of PSI Analysis—Skyscrapers | Statistics (mm) | | |
|---|---|---|---|
| | **Mean/Average** | **Maximum** | **Minimum** |
|  DREAM TOWER (8) | −0.12 | 2.31 | −2.4 |
|  LIMASSOL DEL MAR (5) | −0.36 | 2.51 | −2.23 |
|  THE ICON (10) | 0.29 | 3.58 | −3.61 |

**Table 6.** *Cont.*

| Map of PSI Analysis—Skyscrapers | Statistics (mm) | | |
|---|---|---|---|
| | Mean/Average | Maximum | Minimum |
|  | −0.66 | 2.42 | −4.18 |
|  | 0.02 | 2.58 | −3.41 |
|  | 0.09 | 4.71 | −1.70 |

**Table 6.** *Cont.*

| Map of PSI Analysis—Skyscrapers | Statistics (mm) | | |
|---|---|---|---|
| | Mean/Average | Maximum | Minimum |
|  | −0.16 | 5.23 | −4.00 |
|  | −0.21 | 2.33 | −2.10 |

### 3.1.2. PSI Results Analysis of Tall Buildings on Limassol Coastal Front

Olympic Residence, The Oval, iHome, and Arc Ship Tower are included in the tall buildings category (Table 7). PSI geospatial analysis of the tall buildings reveals that the Olympic Residence building, which follows a two-building complex construction, has an average displacement rate of −1.7 mm in the vertical component. Equally notable is that the specific buildings analyzed had completed construction in the year 2012. The lowest observed minimum value of −4.17 mm among all tall buildings corresponded to Olympic Residence. Concerning the Oval and Arc Ship building, the average displacements of height are equal to −0.41 and −0.42 mm, respectively.

**Table 7.** PSI analysis results of tall buildings on Limassol coastal front.

| PSI Analysis Results of Tall Buildings on Limassol Coastal Front | Statistics (mm) | | |
| --- | --- | --- | --- |
| | Mean/Average | Maximum | Minimum |
| | −1.70 | 2.43 | −4.17 |
| 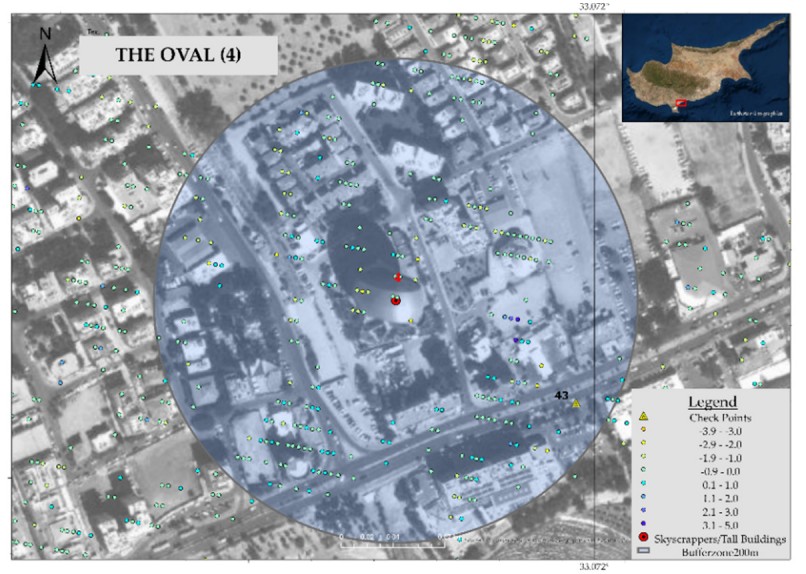 | −0.37 | 3.75 | −2.34 |
| 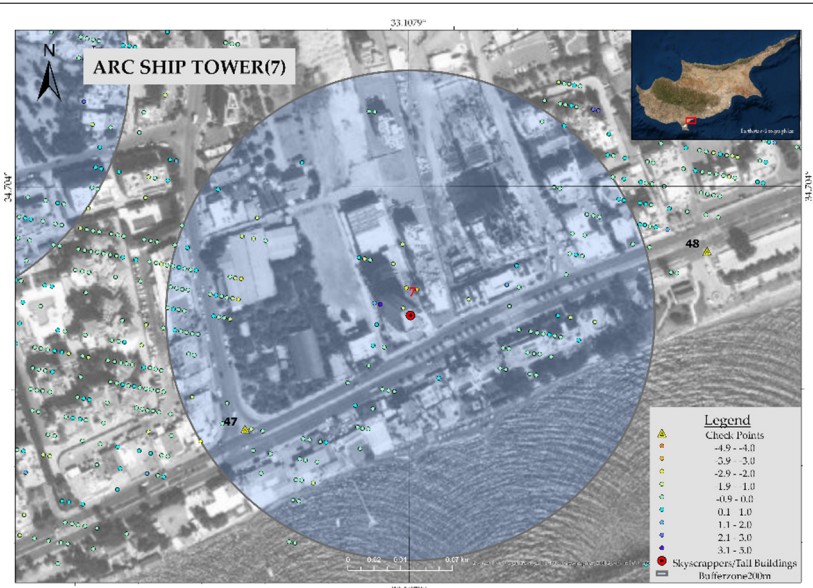 | −0.42 | 1.7 | −3.00 |

*3.2. Final Land Subsidence Results of Limassol Coastal Front according to PSI and Leveling Techniques*

The final results of the study are presented in Table 8. Studying the distribution and the number of the PSs in every buffer zone can be characterized as crucial since it provides information about the sufficiency of the results. The average number of distribution was 182, ranging between 86 (Dream Tower) and 395 (ICON). Concerning the distribution of PSs in the buffer zone of 200 m around each building, this depended on the surrounding area and the construction phase in each case. A similar measurement unit in millimeters was indispensable for the compatibility correctness of the results. Ergo, multiplication by 4 (years of study: 2017–2021) of the value corresponds to PSI mean vertical deformation (mm) computed for the final land subsidence rates for PSI analysis from 2017 to 2021 (mm).

**Table 8.** Final land subsidence results for Limassol coastal front according to PSI and leveling techniques.

| Name | Mean Displacement Rates by PSI Analysis from 2017 to 2021 (mm) | Mean Displacement Rates by Leveling from 2017 to 2021 (mm) | Vertical Displacement Difference from 2017 to 2021 (mm) |
|---|---|---|---|
| The One | −2.68 | −2.500 | −0.18 |
| Trilogy | −2.76 | −2.500 | −0.26 |
| The Dream Tower | −0.48 | −0.102 | −0.291 |
| Limassol Del Mar | −1.44 | −5.003 | −3.563 |
| The Icon | 1.16 | 2.01 | 0.85 |
| Sky Tower | 0.36 | 0.043 | 0.382 |
| Olympic Residence | −3.48 | −3.000 | −0.48 |
| The Oval | −1.48 | −1.238 | −1.439 |
| iHome | −0.84 | −0.484 | −0.662 |
| DTA Tower | −0.054 | −1.901 | −1.847 |
| Arc Ship Tower | −1.68 | −1.290 | −1.485 |
| MARR Tower | 0.080 | 2.090 | 1.010 |

The validation of PSI results was accomplished by leveling measurements. The leveling procedure had a completed accuracy of 7 mm in an almost 7 km length line. For the application of the comparison and validation of two techniques, the nearest benchmark to the corresponding building was selected. This led to a new assumption; the rate of the vertical displacement of leveling benchmark was accordingly compared with the mean vertical displacement of the PSs in their buffer zone. Particularly, the benchmarks with ID LVD041, LVD030, and LVD042 could be compared to the ICON, Trilogy, ONE, and Olympic Residence buildings, respectively. In the buffer zone of the Oval building, the 43 benchmark was included, while the MARR Tower used the LVD033 benchmark. Moreover, the DTA Tower was compared with the nearest benchmark, the LVD045. Opposite the Limassol Del Mar construction the LVD047 benchmark was measured, while the LVD048 benchmark was used for iHome and the Dream Tower.

As it is shown in Table 8, only three cases exceed 2 mm; the One, Trilogy, and Olympic Residence, with vertical land deformation values of −2.393 mm, −5.008 mm, and −4.906 mm, respectively. As seen in the case study described in Table 2, those structures were built next to each other at a distance of one kilometer, with two of them being skyscrapers and one a tall building. Buildings such as the Dream Tower, Sky Tower, iHome, DTA Tower, and MARR Tower have an average value of vertical deformation below 0.5 mm, with the minimum value corresponding to MARR Tower.

## 4. Discussion

In this study, the first assessment of land subsidence on the Limassol coastal front using Copernicus Sentinel-1 mission data from 2017 to 2021, open software toolboxes, and the leveling technique for the comparison and validation of SAR results is rigorously presented. As illustrated in the Results section, newly built structures affect the Limassol coastal front by subsidence and uplift, ranging from almost −5 mm to 5 mm [Figures 9 and 10].

## Land Subsidence (mm)

| | |
|---|---|
| ▭ | -4.83186 - -1.85095 |
| ▭ | -1.85094 - -1.01629 |
| ▭ | -1.01628 - -0.539338 |
| ▭ | -0.539337 - -0.181627 |
| ▭ | -0.181626 - 0.136339 |
| ▭ | 0.13634 - 0.49405 |
| ▭ | 0.494051 - 0.931253 |
| ▭ | 0.931254 - 1.68642 |
| ▭ | 1.68643 - 5.30328 |

**Figure 9.** Legend of the colors used as presented in the final map of results.

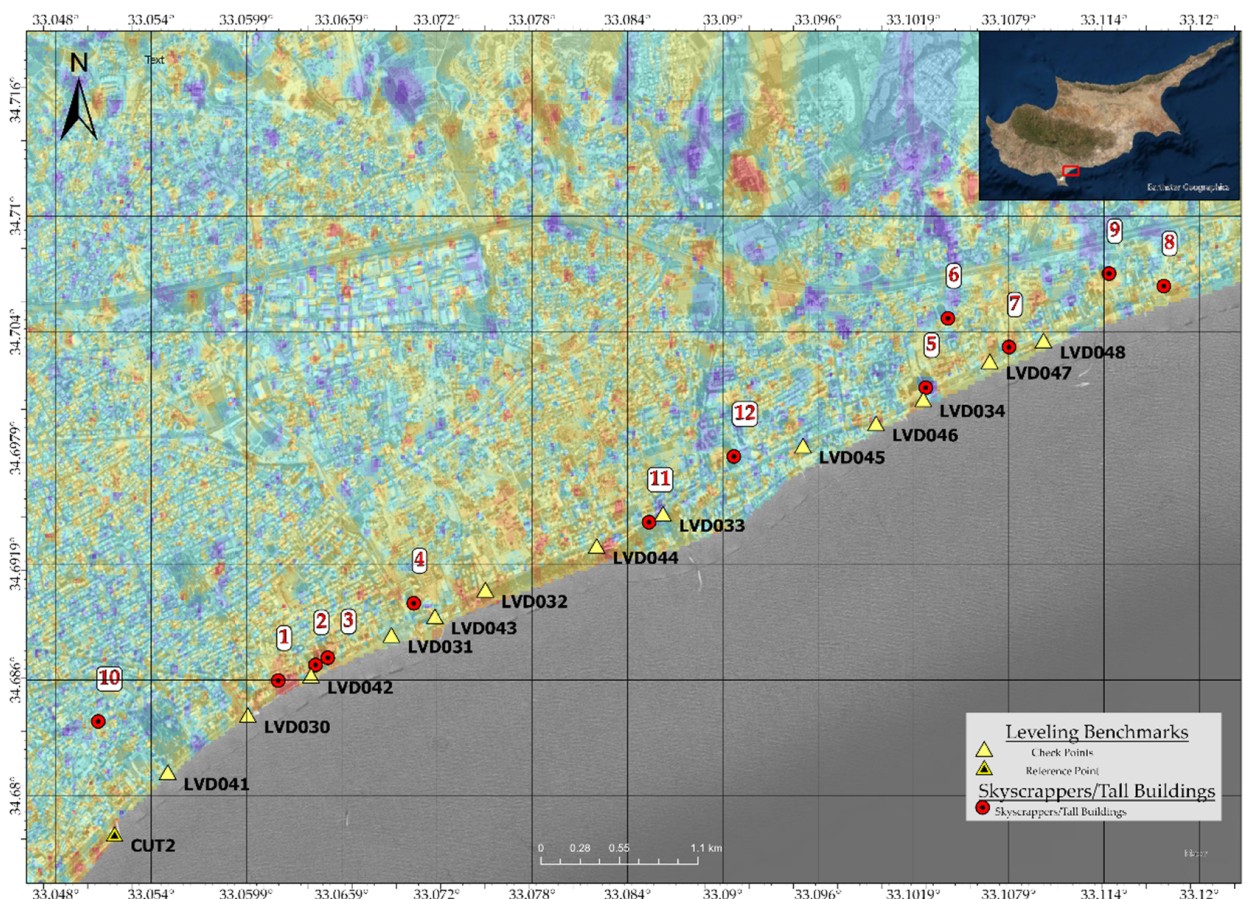

**Figure 10.** Land subsidence on Limassol coastal front using the IDW interpolation.

Due to the medium spatial resolution of Sentinel-1 images, for the estimation of the displacements in the vertical component, the PSs created by the satellite data processing were isolated in a 200 m buffer zone around each building for the current study. The building most affected by land subsidence is a skyscraper named Trilogy, with an average value of −5.008 mm, while the least affected building is the MARR Tower, with a slight uplift of 0.245 mm. Near the region of the Trilogy, the One and Olympic Residence buildings were also built, leading to the classification of the area as high-risk. Similar values of land

subsidence were presented in the Olympic Residence two-building complex at −4.906 mm over five years. Beyond that, subsidence-affected buildings included the Dream Tower, Limassol Del Mar, the Oval, iHome, DTA Tower, and Arc Ship Tower. Contrarily, in some cases, it can be noticed that uplift happens. Those areas are the ICON, Sky Tower, and MARR Tower. Although uplift values were rated as almost zero millimeters over a period five years, the cause of the specific change shall be identified. Concerning being in an area reclaimed from the sea, the deformations may be due to ground settlement taking place in the filler materials under the surface. Positive accumulated ground motion trends can also be shown as relative movements towards the satellite.

Of note is that when studying the behavior of displacements in the environment surrounding artificial objects, it is advisable to explore the whole area. For example, land subsidence on the Limassol coastal front could be looked at regionally instead of buildings as a unit. In that case, it can be clearly seen that the area where those buildings are constructed is affected the most. Furthermore, especially in the Molos area of Limassol, in the west part of the studied area, there are places in which the land subsidence is equal to −5 mm/year. Generally analyzing each location in the coastal plain, it can be deduced that except for the new massive development, the erosion of the coastal plain, which is caused by various factors, should be studied. Various types of land subsidence may occur in the Limassol coastal front, namely subsidence due to groundwater extraction, the mean sea level rise, subsidence induced by the load of the constructions, and geotechnique subsidence caused by the alluvium soil. Additionally, it is possible to highlight the presence of new construction sites created in the area of interest.

Finally, due to the recent construction dates, further research in the study area could be directed to applying new combinations of technologies and methodologies. Although leveling is a time-consuming procedure, additional leveling benchmarks could be installed to add density to the current height network, with every benchmark corresponding to one building. Apart from that, GNSS measurements could be used for the comparison and validation ground-based method for InSAR. A more accurate displacement assessment could show more sufficient results in the following years since the number of satellite images will be extended, and electronic transponders could installed to further investigate Limassol, as it is the perfect city for studying the urban environment through SAR techniques. All the procedures could be automatized, and the possibility of the real-time monitoring of land subsidence could be realized. Through the CyCLOPS (INFRASTRUCTURES/1216/0050) program, corner reflectors have been installed all around Cyprus with promising results.

## 5. Conclusions

The combination of using space- and ground-based data for detecting and monitoring changes in the coastal zone of Limassol after the exponentially rapid growth can be characterized as innovative since no case of detecting changes in urban areas in Cyprus has been studied in the past. This first assessment was accomplished using a combination of techniques such as PSI and leveling. Fifty (50) Sentinel-1 images were used in the PSI process, and thousands of PSs were created. Those PSs were isolated in a 200 m buffer zone, and the average/mean value of the vertical land displacement was computed. For the validation of the PSI results, a leveling procedure was performed, measuring height differences in thirteen (13) benchmarks from 2017 to 2021. Detected displacement using the PSI technique showed land displacement around the majority of the buildings ranging from almost −6 mm to 4 mm over the course of a five-year study. These displacements could be related to coastal construction due to the rapid development growth in the area combined with various atmospheric components. Leveling results have shown similar results with a land displacement range of −3.012 mm to 0.092 mm. Combining those methodologies, the land displacement in the Limassol coastal area was computed between −5 mm and 4 mm from 2017 to 2021. The current research has carried out the beginning stages of monitoring the land displacement locally (Limassol) and regionally (Cyprus).

**Author Contributions:** Conceptualization, C.D.; methodology, K.F. and C.D.; validation, K.F., D.K., M.P., R.B., M.E., D.G.H. and C.D.; formal analysis, K.F., D.K. and C.D.; investigation, K.F., D.K., G.M. and C.D.; resources, C.D.; writing—original draft preparation, K.F.; writing—review and editing, K.F. and C.D.; visualization, K.F.; supervision, C.D.; project administration, C.D.; funding acquisition, C.D. and D.G.H. All authors have read and agreed to the published version of the manuscript.

**Funding:** C.D. and D.G.H. have been financially supported by the 'EXCELSIOR': ERATOSTHENES Excellence Research Centre for Earth Surveillance and Space-based Monitoring of the Environment H2020 Widespread Teaming project (https://excelsior2020.eu, (accessed on 28 December 2021)). The 'EXCELSIOR' project has received funding from the European Union's Horizon 2020 research and innovation programme under Grant Agreement No 857510 and from the Government of the Republic of Cyprus through the Directorate General for the European Programmes, Coordination and Development. K.F. and D.K. have been financially supported by the CyCLOPS (RIF/INFRASTRUCTURES/1216/0050) project (https://cyclops.cy, (accessed on 28 December 2021)), which is co-funded by the European Union Regional Fund and the Republic of Cyprus through the Research and Innovation Foundation in the framework of the RESTART 2016-2020 Programme.

**Institutional Review Board Statement:** Not Applicable.

**Informed Consent Statement:** Not Applicable.

**Data Availability Statement:** The Copernicus Sentinel data [2017–2021]. Retrieved from ASF DAAC [26 March 2021], processed by ESA.

**Acknowledgments:** The authors would like to acknowledge the 'ERATOSTHENES: Excellence Research Centre for Earth Surveillance and Space-Based Monitoring of the Environment-EXCELSIOR' (https://excelsior2020.eu/, (accessed on 28 December 2021)) project that has received funding from the European Union's Horizon 2020 research and innovation programme under grant agreement No 857510 (Call: WIDE-SPREAD-01-2018-2019 Teaming Phase 2) and the Government of the Republic of Cyprus through the Directorate General for European Programmes, Coordination and Development. This paper is under the activities of the 'EXCELSIOR' project. Moreover, the authors would like to acknowledge the support of the 'CyCLOPS' (RIF/INFRASTRUCTURES/1216/0050) project (https://cyclops.cy, (accessed on 28 December 2021)), which is co-funded by the European Regional and Development Fund and the Republic of Cyprus through the Research & Innovation Foundation in the framework of the RESTART 2016–2020 Programme.

**Conflicts of Interest:** The authors declare no conflict of interest.

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
