# Peer review of "Space-Based Displacement Monitoring of Coastal Urban Areas: The Case of Limassol’s Coastal Front"

_remotesensing, doi:10.3390/rs14040914_

Round 1

Reviewer 1 Report

This manuscript include a study about the quantification of the land displacements in the Line of Sight in Limassol (Cyprus) using satellite images. The authors also compute the effect of skycrappers in the subsidence in the coastal lines

In general, the manuscript should be improved in deepth before taking into account for a publication in Remote Sensing. There are two important points to be noted:

1.- The images and tables should necessary be improved (more indications below)

2.- A rigorous statistical analysis with the quality of the results (involving the conclussions) shoud be included. The results analysis are weak and should be amplified and improved

Until the authors make a substantial improvement to the entire article, I believe it should not be considered. Although I will advide the editor as a major revision, even though, it is near a rejection.

Major comments about the tables and figures.

  • Table 1 is not clear, please explain the studies and the location. It is not useful to include the subsidence rate if you do not indicate the situation of the coastal study
  • Figure 1 and Table 2: I suggest to include the location of the skycrapers buildings numbering in figure 1
  • Figure 2: I suggest to put a lower level of blue to a better readness
  • Figure 3: This figure should be remake in deepth. The colours and the text are not clear. I suggest bigger maps with more definition. The caption must be improved
  • Figure 4. More definition would be necessary. The caption must be improved with more explanation. Geographic coordinates latitude and longitude are not described.
  • Figure 1,6 and Figure 5.- The source of the image should be indicated in the caption
  • Figure 6 has the points almost illegible
  • Table 4, 5 and 6 More statistical parameters should be included. The data shown are weak in general

Author Response

Thank you for reviewing our manuscript, in detail.

Response to Reviewer 1 Comments

Space-based Deformation Monitoring of Coastal Urban Areas: The Case of Limassol’s Coastal Front

Point 1: Table 1 is not clear, please explain the studies and the location. It is not useful to include the subsidence rate if you do not indicate the situation of the coastal study

We would like to thanks for your suggestions. As we mention in the lines 60-68 the table describes a brief literature review in previous works located in coastal areas, facing land subsidence issues using Sentinel-1 satellite mission. These works helped us to determine the most suitable technique (PSI/SBAS/DinSAR) for our application. The main goal is to specify the different techniques and the average results of each study, and, not to describe them in detail. Noteworthy is that only a few studies combine ground-based data (e.g. levelling) to validate and compare the spaced-based SAR results. Reading a plethora of previous studies, almost all coastal areas consisted of alluvial soil (including Limassol). We updated the lines 60-68, so the Table 1 is more clear, useful and efficient for the current article.   

Point 2: Figure 1 and Table 2: I suggest to include the location of the skycrapers buildings numbering in figure 1

We added a new column with the building identifiers (according to Figure 1) and updated Table 2 according to your suggestings (by adding the coordinates of each building in WGS84).

Point 3: Figure 2: I suggest to put a lower level of blue to a better readness

Figure 2 has been updated.

Point 4: Figure 3: This figure should be remake in deepth. The colours and the text are not clear. I suggest bigger maps with more definition. The caption must be improved

The figure was replaced with an other figure showing a mean velocity map of the extracted points with coordinates as computed by STAMPS and the caption was updated as well. (see lines 188-192)

Point 5: Figure 4. More definition would be necessary. The caption must be improved with more explanation. Geographic coordinates latitude and longitude are not described.

Figure 4 and its caption have been updated to be more understandable. (see lines 205-213)

Point 6: Figure 1,6 and Figure 5.- The source of the image should be indicated in the caption

There is no source for these images, since they were created by our research team. All images were exported through layouts/maps from ArcGIS Pro using our geodatabase, that includes all the relevant information for the study, generally. Do you suggest mentioning the software at the caption?

Point 7: Figure 6 has the points almost illegible

A change was applied to the basemap of the image, as long as a little upgrade in the size of the labels in the legend was applied.

Point 8: Table 4, 5 and 6 More statistical parameters should be included. The data shown are weak in general

Tables have been updated and explained in the text.

Reviewer 2 Report

Dear authors,

you have written very interesting article according to my opinion, and to my field of interest. English language needs minor improvements. You cited the literature mostly correctly.

The good points of the article are:

  1. interesting topic regarding deformation analysis and comparison of methods.

According to my opinion, the weak points of your article are:

  1. fundamental flow – the comparison of PSI displacements in LoS with precise geometric levelling orthometric heights
  2. leveling network – more detailed explanation is needed
  3. technical shortcomings of the article
  4. references are missing

The improvement should be done in the following:

Ad. 1)

To my opinion this is the weakest point of your article. Is it possible to compare PSI displacements in LoS with precise geometric levelling orthometric heights, i.e., vertical displacement? I couldn’t find did you transform the displacements from PSI LoS to vertical displacements. If you didn’t, please explain how you could compare then this two methods.

Ad. 2)

In section 2.2.3 more detailed leveling network figure is missing. It is not clear where are the reference benchmarks and where are the control benchmarks if they exist. Should be explained to my opinion. I don’t understand if this is the relative or the absolute network, i.e., are the displacements only relative between the benchmarks or are they absolute in relation to some reference benchmarks outside these shown benchmarks.

Also, for better understanding and analysis provided later in the article in section 3.2 I suggest that you put the table in this section 2.2.3 with benchmark heights for every year from 2017 to 2021.

Ad. 3)

There is a lot of technical shortcomings in the article, mostly related to cross references of the figures in the text. Must be corrected.

Figures 5 and 6 needs improvements regarding the contrast of the dots and triangles – detailly in the suggestion by lines.

Units should be written according to SI format, i.e., not 5mm but 5 mm.

Terminology – I suggest that you replace the term deformation with displacement in the whole article. Because, these are not the synonyms and have different meaning.

Ad. 4)

References number 7 and 23 are missing in the text.

Suggestion by “lines”:

- line 2 – I suggest that you replace deformation with displacement since displacements are the value that have been measured by different techniques

- line 23 – please add keyword leveling (because this method is used for validation of the detected deformations)

- line 63 - As shown in Error! Reference source not found – please correct

- line 102 - Error! Reference source not found – please correct

- line 108 – table 2 – I suggest that you removed it because it is irrelevant for the article, it does not contribute to it. I suggest that you just put an URL in the text, where the reader can find this information.

- line 120 - Error! Reference source not found. – please correct

- line 145 - Error! Reference source not found– please correct

- line 182 - Error! Reference source not found. – please correct

- line 198 and 200 - Error! Reference source not found – please correct

- line 202 – I suggest that you change the color of the dots, to e.g. red

- line 205 and 206 - As a ground-based geodetic technique, precise leveling was selected for its simplicity and unmatched accuracy. I agree with this statement, but references are needed here. And I think that you are talking about precise geometric levelling.

- line 207 - [Error! Reference source not found – please correct

- line 213 and 214 - I suggest that you change the color of the triangles, to e.g. red

- line 216 I suggest that you replace industrial grade with high precise…

- line 216 and 217 - [Error! Reference source not found.] – please correct

- line 210 - ±1.0 mm (standard deviation) per 1 km double-run measurements – did you check the accuracy of the instrument according to ISO norm 17123. If yes please wrote this, if not please wrote the manufacturer information about the accuracy and put the reference.

- line 225 – K – is the length of the levelling line

- line 266 - n Error! Reference source not found. and Error! Reference source not found. – please correct

- line 271 - n Error! Reference source not found. and Error! Reference source not found. – please correct

- line 272 - mean, minimum, and ??? value – and maximum value

- line 277 - Error! Reference source not found. and Error! Reference source not found. – please correct

- line 287 – table 4 – for sky tower the maximum value is missing. And I suggest that you wrote the names of the towers above the pictures according to the names in lines 276 and 277

- line 290 - Error! Reference source not found. and Error! Reference source not found. – please correct

- line 292 - average deformation rate up to -1.7 mm/year – this is not deformation rate. This is displacement or shift or subsidence rate. Displacements and deformations are different terms.

- line 301 - Error! Reference source not found. and Error! Reference source not found. – please correct

- line 312 - The leveling procedure has a completed accuracy of 0.01cm. – I agree. But the reference is needed or your own examination according to ISO norm 17123.

- line 317 - Error! Reference source not found. and Error! Reference source not found. – please correct

- line 318 - vertical land deformation – displacement or settlement

- line 320 - Error! Reference source not found. and Error! Reference source not found. – please correct

- line 325 and 326 – table 6 - deformation please replace with displacement or settlement. I suggest with displacements in the whole article. The column Number of PS is irrelevant for this comparison – I suggest that you removed from the table. Instead of column Average Value of Vertical Deformation 2017-2021 (mm) I suggest that you calculate and write the difference between the PSI displacement from 2017 to 2021 and Leveling displacement from 2017 to 2021. This data is relevant for the analysis. I think that Average Value of Vertical Deformation 2017-2021 (mm) that you calculate as mean value of PSI mean displacement and leveling mean displacement doesn’t represent anything, i.e., at the beginning of the article you wrote that you would compare the displacements from PSI with levelling since they are the most accurate and reliable. You should rewrite the discussion also, since you cant refer to this values.

- line 362-373 – I agree with this strongly! It is necessary!

Best regards.

Author Response

Thank you for reviewing our manuscript, in detail.

Response to Reviewer 2 Comments

Space-based Deformation Monitoring of Coastal Urban Areas: The Case of Limassol’s Coastal Front

Point 1:To my opinion this is the weakest point of your article. Is it possible to compare PSI displacements in LoS with precise geometric levelling orthometric heights, i.e., vertical displacement? I couldn’t find did you transform the displacements from PSI LoS to vertical displacements. If you didn’t, please explain how you could compare then this two methods.

We would like to thank you for your valuable comment. This was something that we forgot to mention, indeed. The phase values were converted to displacements values using the SNAP software. According to the bibliography, the displacement values are in LoS and measured in meters. To enable comparison between PSI displacements in LoS with precise geometric levelling orthometric heights, Eq. XX was applied to the displacement values to convert them to vertical displacements. The methodology and the results were updated, as well.  

Point 2: In section 2.2.3 more detailed leveling network figure is missing. It is not clear where are the reference benchmarks and where are the control benchmarks if they exist. Should be explained to my opinion. I don’t understand if this is the relative or the absolute network, i.e., are the displacements only relative between the benchmarks or are they absolute in relation to some reference benchmarks outside these shown benchmarks.

Text, table and figure have been updated. See lines 215-255

Also, for better understanding and analysis provided later in the article in section 3.2 I suggest that you put the table in this section 2.2.3 with benchmark heights for every year from 2017 to 2021.

A table has been created in the section 2.2.3 with benchmark heights for years 2017 and 2021 (see line 227)

Point 3:  There is a lot of technical shortcomings in the article, mostly related to cross references of the figures in the text. Must be corrected.

Figures 5 and 6 needs improvements regarding the contrast of the dots and triangles – detailly in the suggestion by lines.

The cross-references to figures have been updated and corrected.

Units should be written according to SI format, i.e., not 5mm but 5 mm.

Units have been updated.

Terminology – I suggest that you replace the term deformation with displacement in the whole article. Because, these are not the synonyms and have different meaning.

Thank you for your comment. The change has been carried out as you suggested.

Point 4: References number 7 and 23 are missing in the text.

The specific references were updated and inserted in the text.

Suggestion by “lines”:

- line 2 – I suggest that you replace deformation with displacement since displacements are the value that have been measured by different techniques.

The replacement of word deformation with displacement was applied.

- line 23 – please add keyword leveling (because this method is used for validation of the detected deformations)

The keyword leveling has been added.

- line 63 - As shown in Error! Reference source not found – please correct

The error has been fixed.

- line 102 - Error! Reference source not found – please correct

The error has been fixed.

- line 108 – table 2 – I suggest that you removed it because it is irrelevant for the article, it does not contribute to it. I suggest that you just put an URL in the text, where the reader can find this information.

As we mention in the lines 60-68 the table describes a brief literature review in previous works located in coastal areas, facing land subsidence issues using Sentinel-1 satellite mission. These works helped us to find the most suitable technique (PSI/SBAS/DinSAR) for our application. The main goal is to specify the different techniques and the average results of each study, and, not to describe them in detail. Noteworthy is that only a few studies combine ground-based data (e.g. levelling) for validate and compare the spaced-based results. Reading a plethora of previous studies, almost all coastal areas constisted of alluvial soil (including Limassol). Therefore, we kindly think it is worthy of inclusion.

- line 120 - Error! Reference source not found. – please correct

The error has been fixed.

- line 145 - Error! Reference source not found– please correct

The error has been fixed.

- line 182 - Error! Reference source not found. – please correct

The error has been fixed.

- line 198 and 200 - Error! Reference source not found – please correct

The error has been fixed.

 line 202 – I suggest that you change the color of the dots, to e.g. red

The figure has been updated.

- line 205 and 206 - As a ground-based geodetic technique, precise leveling was selected for its simplicity and unmatched accuracy. I agree with this statement, but references are needed here. And I think that you are talking about precise geometric levelling.

A reference has been added to support this statement.

- line 207 - [Error! Reference source not found – please correct

The error has been fixed.

- line 213 and 214 - I suggest that you change the color of the triangles, to e.g. red

The figure has been updated.

- line 216 I suggest that you replace industrial grade with high precise…

The change has been applied.

- line 216 and 217 - [Error! Reference source not found.] – please correct

The error has been fixed.

- line 210 - ±1.0 mm (standard deviation) per 1 km double-run measurements – did you check the accuracy of the instrument according to ISO norm 17123. If yes please wrote this, if not please wrote the manufacturer information about the accuracy and put the reference.

The description of the accuracy of the instrument can be found in the datasheet of the instrument and is defined according to the ISO-17123-2 norm. This statement is added to the text.

- line 225 – K – is the length of the levelling line

The description of the K was added to the text. (Line 250)

- line 266 - n Error! Reference source not found. and Error! Reference source not found. – please correct

The error has been fixed.

- line 271 - n Error! Reference source not found. and Error! Reference source not found. – please correct

The error has been fixed.

- line 272 - mean, minimum, and ??? value – and maximum value

The statistical value maximum has been added to the text.

- line 277 - Error! Reference source not found. and Error! Reference source not found. – please correct

The error has been fixed.

- line 287 – table 4 – for sky tower the maximum value is missing. And I suggest that you wrote the names of the towers above the pictures according to the names in lines 276 and 277

The maximum value was added and the table 6 was updated.

- line 290 - Error! Reference source not found. and Error! Reference source not found. – please correct

The error has been fixed.

- line 292 - average deformation rate up to -1.7 mm/year – this is not deformation rate. This is displacement or shift or subsidence rate. Displacements and deformations are different terms.

Thank you for your comment. The difference is understandable, the terms were updated.

- line 301 - Error! Reference source not found. and Error! Reference source not found. – please correct

The error has been fixed.

- line 312 - The leveling procedure has a completed accuracy of 0.01cm. – I agree. But the reference is needed or your own examination according to ISO norm 17123.

As mentioned earlier, the accuracy standard is the ISO17123-2 (according to the manufacturer datasheet). There was a typing error at this stage. The levelling accuracy was 7mm (0.7cm) not 0.01cm.

- line 317 - Error! Reference source not found. and Error! Reference source not found. – please correct

The error has been fixed.

- line 318 - vertical land deformation – displacement or settlement

The error has been fixed.

- line 320 - Error! Reference source not found. and Error! Reference source not found. – please correct

The error has been fixed.

- line 325 and 326 – table 6 - deformation please replace with displacement or settlement. I suggest with displacements in the whole article. The column Number of PS is irrelevant for this comparison – I suggest that you removed from the table. Instead of column Average Value of Vertical Deformation 2017-2021 (mm) I suggest that you calculate and write the difference between the PSI displacement from 2017 to 2021 and Leveling displacement from 2017 to 2021. This data is relevant for the analysis. I think that Average Value of Vertical Deformation 2017-2021 (mm) that you calculate as mean value of PSI mean displacement and leveling mean displacement doesn’t represent anything, i.e., at the beginning of the article you wrote that you would compare the displacements from PSI with levelling since they are the most accurate and reliable. You should rewrite the discussion also, since you cant refer to this values.

The tables were updated and the correct vertical displacements were computed. Thank you for the information. Additional literature review has been studied and the results now are making sense and are totally comparable with the leveling procedure. See Section Results and Discussion.

Round 2

Reviewer 2 Report

Dear authors,

thank you for accepting my suggestions!

The article has been significantly improved.

Best regards.